# A super liquid-repellent hierarchical porous membrane for enhanced membrane distillation

Youmin Hou[1,2], Prexa Shah[1], Vassilios Constantoudis[3], Evangelos Gogolides[3], Michael Kappl [1] ✉ & Hans-Jürgen Butt [1]

Membrane distillation (MD) is an emerging desalination technology that exploits phase change to separate water vapor from saline based on low-grade energy. As MD membranes come into contact with saline for days or weeks during desalination, membrane pores have to be sufficiently small (typically <0.2 μm) to avoid saline wetting into the membrane. However, in order to achieve high distillation flux, the pore size should be large enough to maximize transmembrane vapor transfer. These conflicting requirements of pore geometry pose a challenge to membrane design and currently hinder broader applications of MD. To address this fundamental challenge, we developed a super liquid-repellent membrane with hierarchical porous structures by coating a polysiloxane nanofilament network on a commercial micro-porous polyethersulfone membrane matrix. The fluorine-free nanofilament coating effectively prevents membrane wetting under high hydrostatic pressure (>11.5 bar) without compromising vapor transport. With large inner micro-porous structures, the nanofilament-coated membrane improves the distillation flux by up to 60% over the widely used commercially available membranes, while showing excellent salt rejection and operating stability. Our approach will allow the fabrication of high-performance composite membranes with multi-scale porous structures that have wide-ranging applications beyond desalination, such as in cleaning wastewater.

Water scarcity currently affects every continent and ~3 billion people around the world and is one of the greatest challenges in this century. For this reason, efficient desalination methods, that is the separation of fresh water from saline or contaminated water, are required. Membrane distillation (MD) has recently gained much attention given its simple separation mechanism operating at low temperatures and pressure[1–9]. Desalination by the MD process is based on the use of hydrophobic membranes, which contact a heated saline water (normally at 50 to 80 °C) at the feed side. Driven by the temperature difference across the membrane, water evaporates at the membrane-saline interface, diffuses through the pores of the membrane and condenses on the opposite side (normally at ~20 °C), as shown in Fig. 1a. Due to its intrinsic water repellency, the hydrophobic membrane prevents the saline water from passing through while allowing for vapor transport. In this way it separates the volatile (i.e., water) and nonvolatile species (i.e., salts) in the hot saline. As a phase-change-based desalination process, MD is highly suitable to desalinate or concentrate brines above the salinity limit of reverse osmosis (~80 g kg⁻¹). Because MD has a theoretical 100% rejection to non-volatile species and is less sensitive to feed concentration, it is

[1]Max Planck Institute for Polymer Research, Ackermannweg 10, 55128 Mainz, Germany. [2]School of Power and Mechanical Engineering, Wuhan University, 430072 Wuhan, China. [3]Institute of Nanoscience and Nanotechnology NCSR Demokritos, 15341 Agia Paraskevi, Greece. ✉e-mail: kappl@mpip-mainz.mpg.de

**Fig. 1 | Superhydrophobic hierarchical porous PES membrane. a** Schematic of the hierarchical porous structure of nanofilament-coated membrane, which can concurrently enhance the wetting resistance and vapor permeability. **b**–**e** Scanning electron microscopy (SEM) top view images of membranes (**b**, **c**) pristine PES-8 membrane, (**d**, **e**) nanofilament-coated PES-8 membrane at different magnifications. **f**, **g** Cross-section SEM images of the nanofilament-coated PES-8 membrane with hierarchical porous structures. Yellow arrows in (**f**, **g**) denote the nano-porous outer layer on top of micro-porous structures. Red dashed lines represent one of the micro-porous paths inside the membrane. **h** Pore size distribution of the nanofilaments coating, pristine PES-8, PE-0.2 and PTFE-0.2 membranes.

considered a promising method to achieve zero liquid discharge when combined with crystallization technology[10,11]. MD desalination units have a relatively simple configuration that does not require expensive materials or high-pressure process equipment, thus providing easy access to drinking water for people in rural and remote areas[12–14].

Despite intensive efforts, widespread adoption of MD is still hindered by the lack of suitable hydrophobic membranes with durable high wetting resistance and distillation flux[15,16]. Theoretically, distillation flux $J$ of a MD membrane can be expressed as a function of the membrane properties and partial vapor pressure difference ($\triangle p$) across the membrane:

$$J \sim \frac{\varepsilon \mathcal{D}_m}{\tau \sigma RT \left(1 + \frac{\mathcal{D}_m}{\mathcal{D}_K}\right)} \triangle p \qquad (1)$$

where $\varepsilon$, $\tau$, and $\sigma$ are the membrane porosity, tortuosity, and thickness respectively; $R$ is the universal gas constant; $T$ is the mean membrane temperature; $\mathcal{D}_m$ is the molecular diffusion coefficient of vapor in air; $\mathcal{D}_K$ is the Knudsen diffusion coefficient of vapor inside the membrane. At a fixed temperature, the ratio of the diffusion coefficients ($\mathcal{D}_m/\mathcal{D}_K$) is inversely proportional to the nominal pore diameter ($d_n$) of the membrane[4,17]. Therefore, a porous membrane with large pores and higher porosity will maximize the distillation flux.

Durable MD desalination with a high salt rejection necessitates excellent membrane wetting resistance, in order to prevent wetting of salty water into the membrane. The wetting resistance of a membrane is usually assessed quantitatively by the liquid entry pressure (LEP), which is defined as the minimum required pressure for liquid solution

entering the membrane pores. In a simple form based on the Young-Laplace equation[18,19], it can be calculated as LEP $\sim (-4\gamma \cos \theta_Y)/d_{max}$, where $d_{max}$ is the maximum membrane pore diameter, $\gamma$ is liquid surface tension, and $\theta_Y$ is the intrinsic contact angle of the membrane material. Accordingly, membranes with small pore size, narrow pore size distribution, and low surface energy typically show high LEP and excellent salt rejection.

These requirements of both high distillation flux and LEP for MD membranes pose a critical challenge when designing a membrane – large membrane pore size allows efficient distillation, and yet it inevitably increases the susceptibility to liquid penetration. To ensure stable desalination without risk of wetting, the nominal pore diameters of the widely used commercially available membranes are typically less than 0.2 μm, which greatly reduce desalination efficiency. In order to balance the conflicting requirements of distillation flux and LEP, conceptual designs of composite membranes with different pore sizes have been proposed over the past years[4,20]. To date, however, it still remains a challenge to produce a scalable polymeric membrane that can simultaneously enhance distillation flux and wetting resistance via a simple processing method.

In this study, we have developed a fluorine-free superhydrophobic membrane by coating a thin layer of nanofilament network onto the top of a micro-porous membrane matrix, which combines the advantages of multi-scale porous structures (Fig. 1a). This hierarchical topography greatly enhances the LEP of the membrane while retaining a high vapor transfer rate. During the one-week direct contact and air gap membrane distillation tests, the nanofilament-coated membrane demonstrated ~30 to 60% higher

distillation flux than that of the commercially available membranes. Concurrently, the thermal efficiency of desalination was improved from 84% to 93%. Owing to its non-toxic hydrophobic nature and potential for scalable manufacturing, this advanced composite membrane offers an avenue to affordable clean water for the off-grid communities by using low-grade energy.

## Results

### Nanofilament-coated membranes with hierarchical structure

To fabricate the high-performance composite membrane required by MD desalination, we coated a superhydrophobic nano-porous layer onto a micro-porous polyethersulfone (PES) membrane. This commercially available 130-μm-thick PES membrane with nominal pore diameter ($d_n$) of 8 μm (in the following denoted as PES-8) (Fig. 1b, c) acts as a robust supporting framework due to its toughness, good thermal resistance, and relatively high porosity (~75 to 80%). To create the nano-porous layer, the membrane ($64 \times 75$ mm$^2$) was immersed in a mixture of $n$-heptane and toluene (volumetric ratio 1:1), which contained trichloromethylsilane (TCMS) and trace amounts of water (150 ppm)[21,22]. The hydrolyzed trichloromethylsilane in solvent reacted with the hydroxyl groups on the membrane surface and self-assembled into a porous network of polysiloxane nanofilaments. The nanofilaments covered the whole outer surface of the PES membrane, including the large opening pores (Fig. 1d, e and Supplementary Fig. 2). The interwoven structure of nanofilaments resulted in an overhanging morphology with an inward curvature. Due to the exposed methyl groups, the nanofilaments have a low surface energy. This combination of overhang structural topography with low surface energy renders the nanofilament network stable and superhydrophobic, even without using any fluorine-containing reagents[23–27].

To explore the effect of nanofilament coating on MD performance in this study, we selected the commercial polyethylene (PE) and polytetrafluoroethylene (PTFE) micro-porous membranes as the benchmarks for comparison (see properties of all tested membranes in Table 1, the surface morphology of PE and PTFE membranes are shown in Supplementary Fig. 1).

SEM images of cross-sections (Fig. 1f, g) further elaborated the dual-layer topography of the nanofilament-coated PES membrane with multi-scale pore sizes. The coated nanofilaments spontaneously intertwined and formed a dense nano-porous layer on top of the micro-porous membrane matrix (highlighted by arrows in Fig. 1g). At the same time, the nanofilaments also grew inside the membranes and converted the interior surface into a superhydrophobic surface (Fig. 1f, 1g and Supplementary Fig. 3). The inner coating, however, was sufficiently thin so that the geometry of the inner micro-pores remained almost unchanged, offering a high membrane permeability for water

vapor transport. The superhydrophobic nature of inner micro-pores also increases the water nucleation energy barrier, thus resisting the capillary condensation and consequently pore wetting inside the membrane[28].

To analyze the effective pore diameter of a nanofilament network coating, we evaluated the geometrical features based on the computer analysis of SEM images. The details of the analysis process are described in Supplementary Fig. 4 and Supplementary Note 3. The analysis algorithm was first validated through measurements of a pristine PES-8 membrane, commercial PE and PTFE membranes with nominal pore diameters of 0.2 μm (denoted as PE-0.2 and PTFE-0.2, respectively). From the image analysis, the obtained mean pore diameters of the PE-0.2, PTFE-0.2, and PES-8 membranes closely matched the product data, confirming the reliability of the measurements. The computer image analysis revealed that the pore diameter of the nanofilament coating was distributed between 10 to 100 nm. Compared to the commercial membranes, the nano-porous coating resulted in not only a smaller pore size, but a nanoscopic overhanging structure (see Supplementary Fig. 3), which would substantially increase the liquid entry pressure of coated membranes according to the Young-Laplace equation.

### Liquid entry pressure and gas permeability

Desalination performance using the MD process relies essentially on the liquid entry pressure (LEP) and gas permeability of the adopted membrane: the LEP highly affects the salt rejection and the gas permeability dominates the distillate flux. To quantitatively explore the improved wetting resistance of hierarchical membranes, we benchmarked the LEP of nanofilament-coated PES membranes against commercial PE membranes, PTFE membranes, and PES membranes that were rendered hydrophobic by fluorination. The testing was carried out in a custom-made setup which can ramp up the transmembrane pressure difference (see Methods and Supplementary Note 5). For convenient description here, NF-PES-0.1 to NF-PES-8 are used to denote the nanofilament-coated PES membranes with nominal pore diameter ($d_n$) ranging from 0.1 to 8 μm; PE-0.2 to PE-2.5 are used to denote the PE membranes with $d_n$ ranging from 0.2 to 2.5 μm; PTFE-0.1 to PTFE-1 used to denote PTFE membranes with $d_n$ ranging from 0.1 to 1 μm.

For the membranes with single-scale porous geometry (e.g., PE, PTFE, and fluorinated PES membranes), the LEP values dropped sharply with increasing nominal pore diameter $d_n$ (Fig. 2a), which agrees well with the estimation of the Young-Laplace equation when regarding $d_{max} = d_n$. In a practical MD process, the transmembrane pressure difference normally ranges between 0.3 to 1.5 bar according to different operating pressures on the side where permeate is collected[29,30].

## Table 1 | Characteristics of commercial membranes and nanofilament-coated PES membranes

| ID | Material | Nominal pore diameter (μm) | Liquid entry pressure (bar) | Receding contact angle (°) | Contact angle hysteresis (°) |
|---|---|---|---|---|---|
| PE-0.2 | PE | 0.2 | 6 ± 1.1 | 88 ± 3 | 35 ± 3 |
| PE-0.5 | | 0.5 | 1.65 ± 0.4 | 92 ± 2 | 41 ± 3 |
| PE-0.9 | | 0.9 | 1.07 ± 0.12 | 92 ± 3 | 32 ± 4 |
| PE-1.5 | | 1.5 | 0.45 ± 0.1 | 85 ± 3 | 35 ± 3 |
| PE-2.5 | | 2.5 | 0.4 ± 0.1 | 86 ± 2 | 39 ± 4 |
| PTFE-0.1 | PTFE | 0.1 | 6.5 ± 0.55 | 130 ± 2 | 19 ± 3 |
| PTFE-0.2 | | 0.2 | 5.05 ± 0.7 | 132 ± 1 | 17 ± 2 |
| NF-PES-0.1 | Nanofilament-coated PES | 0.1 | >11.5 | 161 ± 3 | 2 ± 1 |
| NF-PES-1.2 | | 1.2 | >11.5 | 159 ± 3 | 3 ± 1 |
| NF-PES-3 | | 3 | 6.5 ± 1.5 | 157 ± 2 | 4 ± 1 |
| NF-PES-5 | | 5 | 5.8 ± 1 | 158 ± 3 | 2 ± 1 |
| NF-PES-8 | | 8 | 5.5 ± 0.8 | 160 ± 4 | 3 ± 2 |

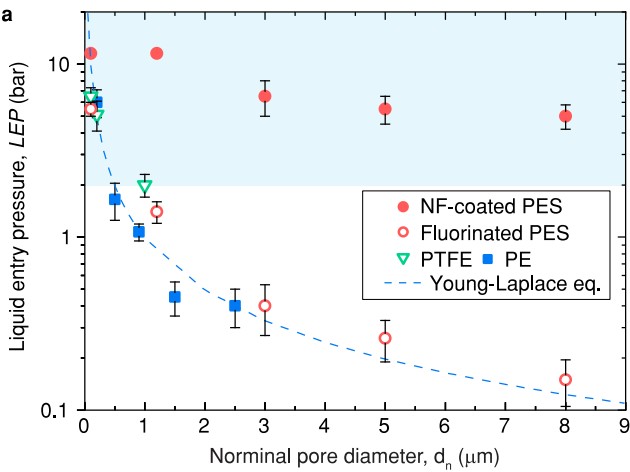

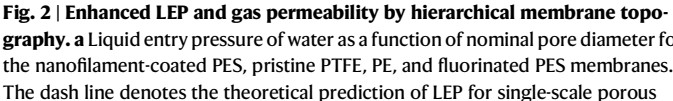

**Fig. 2 | Enhanced LEP and gas permeability by hierarchical membrane topography. a** Liquid entry pressure of water as a function of nominal pore diameter for the nanofilament-coated PES, pristine PTFE, PE, and fluorinated PES membranes. The dash line denotes the theoretical prediction of LEP for single-scale porous membranes from Young-Laplace equation. Error bars indicate standard deviation ($n = 6$). **b** Nitrogen gas permeation flux as a function of transmembrane pressure difference for nanofilament-coated PES-8, pristine PTFE-0.2, PE-0.2, and PES-8 membranes. Error bars indicate standard deviation (n = 6).

To ensure the sufficient safety margin of LEP, the $d_n$ of MD membranes in pilot scale testing is typically between 0.1 and 0.5 μm[30–32].

We found a greatly enhanced LEP for nanofilament-coated PES membranes, as the nano-porous outer layer withstands high capillary pressure. Note that NF-PES-0.1 and NF-PES-1.2 membranes exhibited an extremely high LEP, which even exceeded the limit of our testing setup (11.5 bar). However, when the pore size of PES membranes is considerably larger (e.g., $d_n > 3$ μm), the LEP of the nanofilament-coated membrane gradually goes down with the increasing pore size of matrix. The decline of LEP can be related to the imperfect growth of the nano-porous layer on the big membrane pores as it is difficult to cover the large openings completely with a nanofilament network. Nevertheless, the effective pore diameter of the NF-PES-3, NF-PES-5, and NF-PES-8 membranes were still smaller than 0.3 μm according to the theoretical estimation of the Young-Laplace model. Compared to the fluorinated PES membrane with the same pore diameter, the nanofilament-coated PES membranes raised the LEP by at least 16-fold. Considering the hydraulic pressure in the MD system, the high LEPs of our nanofilament-coated PES membranes ensure a wide applicability in desalination, even in the harsh conditions for treating saline with low-surface-tension liquids (The characteristics of commercial membranes and nanofilament-coated PES membranes are listed in Table 1). Besides membrane wetting by imbibition, as characterized by LEP and described theoretically by the Young-Laplace equation, capillary condensation inside of membrane pores of could in principle be another mechanism of wetting. Such a wetting process could be described theoretically by a grand potential approach for hydrophilic surfaces[33]. However, the micrometer-sized pores of the hydrophilic core PES membrane are too large to lead to capillary condensation. Due to the hydrophobic character of the nanofilaments, capillary condensation is not expected to occur within the coating layer. If capillary condensation would occur within the nanofilament layer, it would lead to a gradual decrease of the distillation flux, which we do not observe even during prolonged distillation tests.

To evaluate the membrane resistance to vapor transport, we characterized the gas permeation of PE, PTFE, and nanofilament-coated PES membranes under differing transmembrane pressures (see Methods and Section 6 in Supplementary Information). A linear dependence between gas permeation flux and transmembrane pressure difference was observed for both single-scale and multi-scale porous membranes (Fig. 2b). After coating with nanofilaments, the gas permeation flux of the NF-PES-8 membrane decreased by ~55% when compared to the uncoated one. Although the mass transfer barrier induced by the nano-porous outer layer is obvious, the large micro-porous paths inside the PES-8 membrane guarantee a higher overall gas permeability than the conventional commercial MD membranes. The measurements demonstrated that the gas permeation flux of NF-PES-8 membrane was nearly 10 times above that of a commercial PE-0.2 membrane and ~3 times higher than PTFE-0.2 membrane. Given the enhanced LEP shown above, the nanofilament-coated PES membranes successfully resolve the conflict in MD membrane design[4,34]. The concurrent enhancements of gas permeability and liquid wetting resistance demonstrate the superior performance of our nanofilament-coated membrane compared to those reported previously[35–38].

## Durability

During MD desalination, membranes need to remain in contact with hot water for days, which may affect the physical properties and surface chemistry of membranes[39,40]. To evaluate possible degradation of membrane surfaces, PE, PTFE and nanofilament-coated PES membranes were immersed in Milli-Q water at 80 °C for 1 to 168 h (one week). After drying the tested membranes under a nitrogen stream, we measured the apparent receding contact angle $\theta_r^{app}$ for water and the contact angle hysteresis $\theta_{CAH}$ on the membrane surfaces (Fig. 3a, b). Figure 3c shows the sessile droplet deposited on different membrane surfaces before and after the one-week immersion test. Compared to the great change in droplet morphology on the PE and PTFE membranes, the droplet wetting state on the nanofilament-coated surface remained almost unchanged. During the 168 h immersion test, the NF-PES-8 membrane maintained the super liquid-repellency with $\theta_r^{app}$ greater than 150° and $\theta_{CAH}$ less than 5°. The snapshot in Fig. 3d presents the droplet mobility on a NF-PES-8 membrane before and after the immersion test, respectively. Even when the NF-PES-8 membrane was immersed in hot water for a long time, water droplets still rolled off the surface rapidly, as was the case before the immersion test.

In contrast, the commercial PE and PTFE membranes showed significant deterioration after being immersed in hot water for one week. $\theta_r^{app}$ on PE membranes declined from ~99° to ~16° and $\theta_{CAH}$ increased from ~37° to ~83°. Although PTFE is widely considered to be thermally stable, $\theta_r^{app}$ on the PTFE membranes decreased from ~116° to ~24° and $\theta_{CAH}$ increased from ~25° to ~91°, indicating the considerable loss of liquid-repellency on the surface. A similar decrease in liquid repellency was also observed for a bulk PTFE sample (Supplementary Fig. 11). As shown in Fig. 3e, the water droplet was repelled on the original hydrophobic PE membrane, but it easily wetted and stained the membrane that was immersed in hot water.

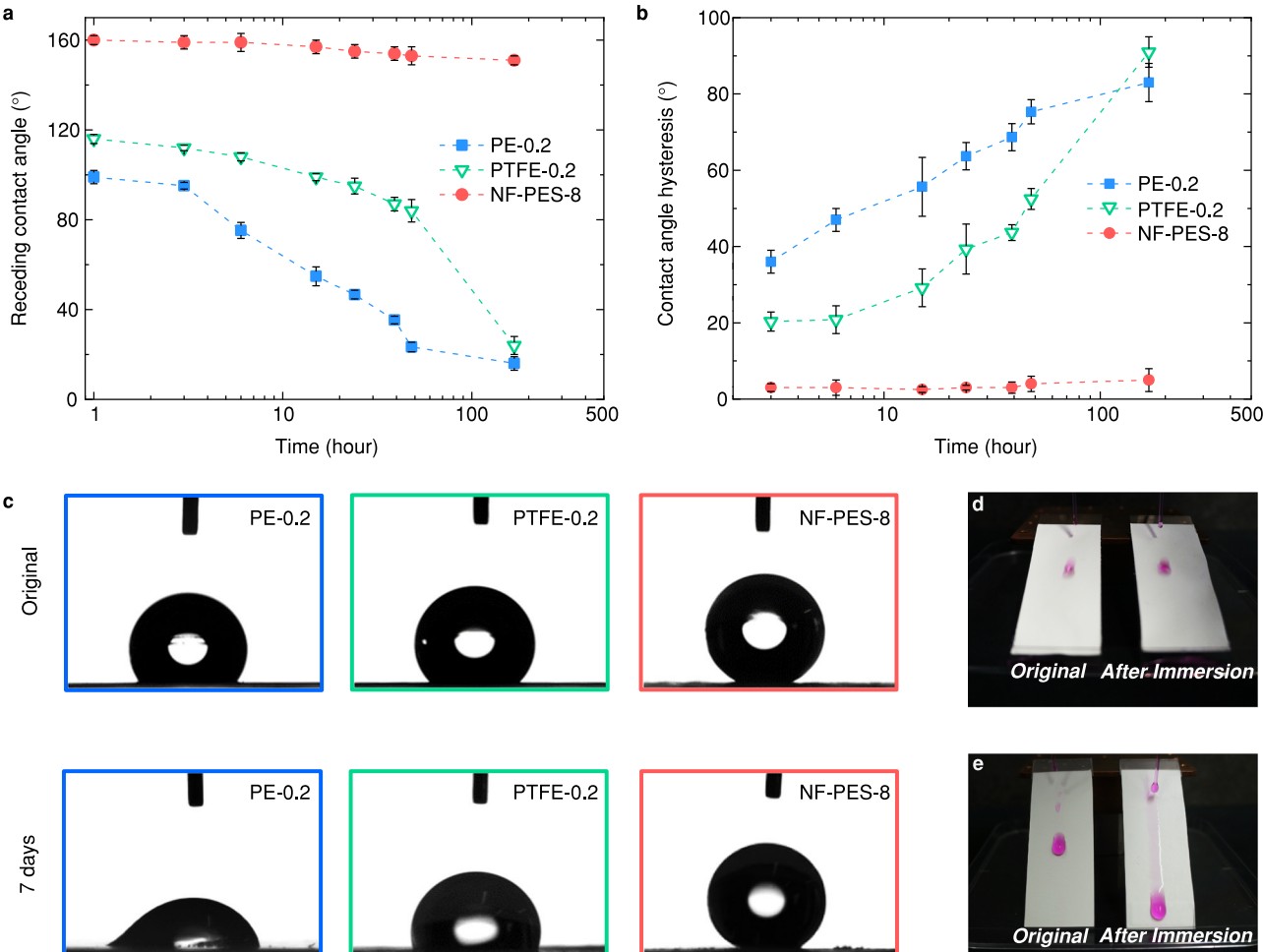

**Fig. 3 | Durability of liquid repellency for the nanofilament-coated membrane.** **a** Receding contact angle $\theta_r^{app}$ and **b** contact angle hysteresis $\theta_{CAH}$ of water on the PE, PTFE and nanofilament-coated PES membranes as a function of immersion time in Milli-Q water at 80 °C. Error bars indicate standard deviation ($n = 6$). **c** Selected snapshots showing the contact angles of a 5 μl water droplet on PE-0.2, PTFE-0.2, and NF-PES-8 membranes before and after 7-day immersion test. The images and contact angle measurements were obtained by using micro-goniometer (Data-Physics OCA35). Snapshots comparing the mobility of dyed water droplets before and after immersion test on (**d**) NF-PES-8 membrane (motion blur due to fast drop movement) and (**e**) PE-0.2 membrane.

We assume that the PE membrane degraded due to an accelerated polymer oxidation in hot water[39,41], which generates polar groups on the PE surface and thus increases the surface energy. The increasing hydrophilicity on PTFE membranes is most likely caused by the change in surface structure and polymer crystallinity at elevated temperatures, based on previous studies of PTFE polymer and membranes[42,43]. In the case of commercial micro-porous membranes, the loss of surface hydrophobicity could cause a gradual wetting of liquid into membranes, decrease desalination efficiency, and even cause the contamination of the distillate by salty water.

The durable superhydrophobicity of NF-PES-8 membranes indicates that by reducing the wetted area on membrane surface, we have an effective way of retarding the polymer degradation in hot water. We estimated the fraction of the water contact area on membrane surfaces by analyzing the receding contact angle. For superhydrophobic arrays of cylindrical micropillars, the apparent receding contact angle ($\theta_r^{app}$) can be expressed as[44],

$$\cos\frac{\theta_r^{app}}{2} \approx (\pi\varphi_{ls})^{1/2} \sin\theta_r \qquad (2)$$

Here, $\theta_r$ is the receding contact angle for water on a flat surface of same material as the membrane. The wetted area $\varphi_{ls} = A_{ls}/A_w$ is the ratio of the projected area of liquid-solid interface ($A_{ls}$) to the projected area of total wetted region ($A_w$), and $\varphi_{lv} = 1 - \varphi_{ls}$ is the fractional area of the liquid-vapor interface. Note that Eq. (2) will only allow a rough estimate since the nanofilament structure is very different from that of cylindrical micropillar arrays. By adopting the specific geometrical model in Eq. 2, the wetted area fractions on PE and NF-PES-8 membranes were estimated to be ~20% and ~1.5% at the beginning of the immersion test, respectively. An alternative approach is to take the liquid droplet in global thermodynamic equilibrium and apply the Cassie-Baxter equation to estimate the surface fraction $\varphi_{ls}$. In this case, the predicted $\varphi_{ls}$ for PE and NF-PES-8 membranes were ~40% and ~5%, respectively. Whatever the approach, the extremely small wetted area $\varphi_{ls}$ on the NF-PES-8 membrane (~1.5 to 5%) explains its long-term thermal stability in the immersion test, and also implies a higher thermal efficiency in desalination. This is because the heat loss by conduction between hot water and the membrane is greatly suppressed[45]. The nanofilament coating also increases the water evaporation area on the membrane surface due to its superhydrophobic nature. As indicated by the measured $\theta_r^{app}$, the NF-PES-8 membrane kept a large fractional area of liquid-vapor interface with $\varphi_{lv} = $ ~98.5% throughout the 48 h test. In the theoretical analysis of the MD process, $\varphi_{lv}$ is practically equivalent to the surface porosity ($\varepsilon$). Therefore, from Eq. 1 distillation flux ($J$) is proportional to $\varphi_{lv}$, i.e., the large $\varphi_{lv}$ on the NF-PES-8 membrane would lead to a substantial increase in the overall production of distillate.

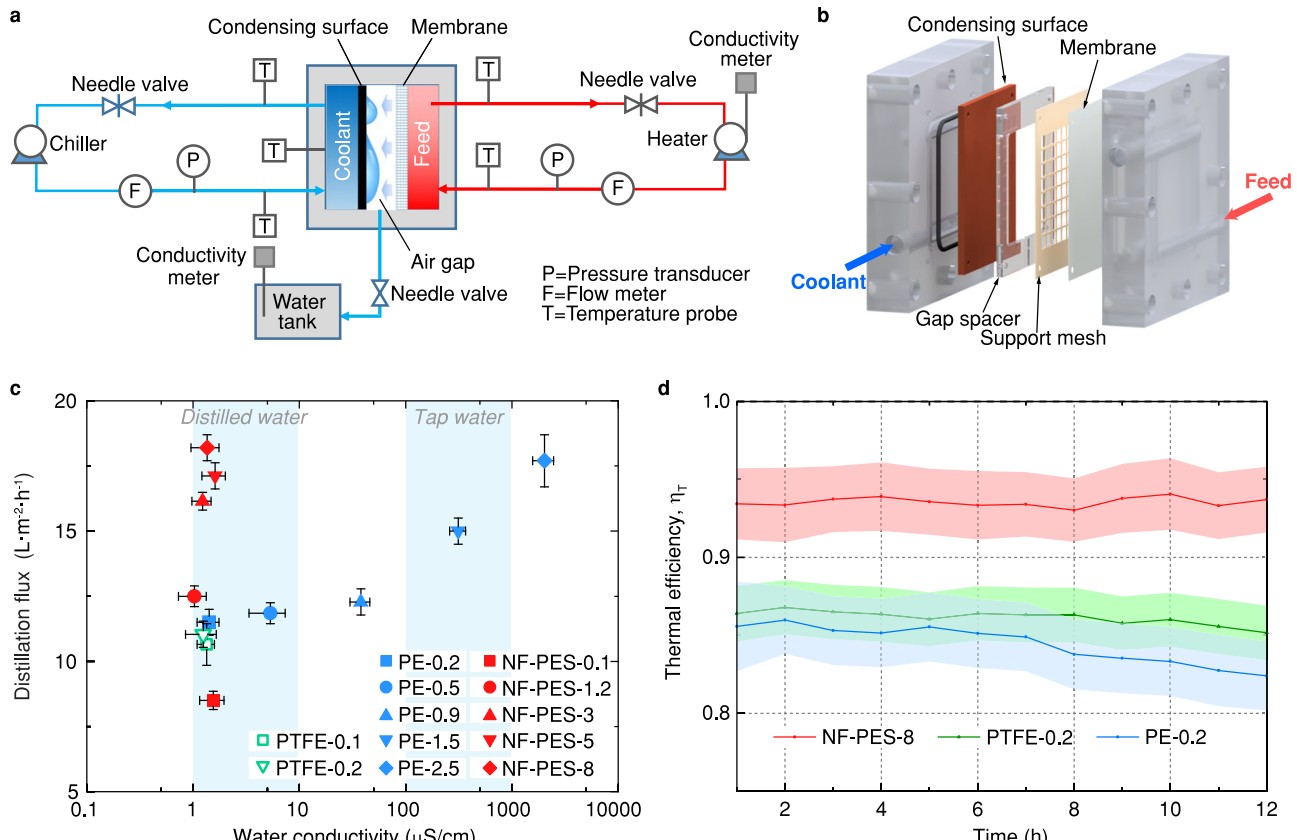

**Fig. 4 | Desalination performance of the nanofilament-coated membranes.**
**a** Schematic of the air gap membrane distillation (AGMD) testing system. **b** 3D schematic depicting the design of the AGMD module. A membrane with an effective area of 19 cm² was mounted between the hot feed flow channel and condensation chamber. An acrylic spacer is inserted to create the required air gap. The support mesh helps to hold the membrane in a planar shape and reduce the membrane deformation due to transmembrane pressure difference.
**c** Experimental steady-state distillation flux as a function of distillate conductivity

for the nanofilament-coated PES, commercial PE and PTFE membranes in 12 h AGMD desalination tests. Temperatures of feed and cooling water were 80 and 20 °C, respectively. Flow rates of feed and cooling water were 1.5 and 2 L min⁻¹, respectively. Error bars indicate standard deviation ($n = 5$). **d** Time evolution of thermal efficiency for the NF-PES-8, PE-0.2, and PTFE-0.2 membranes in 12 h AGMD desalination. The operating conditions are the same as **c**. The error bands indicate the propagation of error associated with the fluid inlet and outlet temperatures, flow rate, and distillate weight measurement.

## Enhanced desalination via hierarchical porous membranes

To highlight the technical potential of nanofilament-coated PES membranes in water desalination, we conducted MD experiments in a custom-made air gap membrane distillation (AGMD) system (Fig. 4a, b, see details in Methods section and Supplementary Note 7). As shown in the schematic drawing of the AGMD module, an air gap was introduced between the membrane and condensing surface, which prevented the membrane from coming into direct contact with the condensed water. The AGMD configuration reduce the conductive heat loss through the membrane, as well as permitting internal latent heat recovery when water condenses on the cooling surface[7]. Owing to the improved thermal efficiency resulting from the air gap and heat recovery, the AGMD process has recently been considered as the first choice for pilot scale testing and future industrial applications[1,46]. In AGMD experiments, the nanofilament-coated PES membranes were tested at different feed water temperatures for more than 12 hours. The conductivity of saline feed water $\sigma_f$ in the MD test was stabilized at the level of seawater ($\sigma_f = 54.0 \pm 0.5$ mS cm⁻¹ at 25 °C), corresponding to about 35 g L⁻¹ of NaCl (99.7% purity). The weight and conductivity of distillate $\sigma_d$ were continuously monitored over time to characterize the distillation flux and salt rejection rate in the desalination process. For comparison, MD performance of commercial PE and PTFE membranes were assessed as being the benchmark.

We tested nanofilament-coated PES membranes in AGMD experiments at a feed temperature of 80 °C. We calculated the salt rejection rate by $(1 - \sigma_d/\sigma_f) \times 100\%$, where $\sigma_d$ and $\sigma_f$ are conductivity

of distillate and feed, respectively. As commonly used membranes in the MD process, hydrophobic PE and PTFE membranes were able to remove more than 99.9% salt from the feed saline (i.e., $\sigma_d < 54$ mS cm⁻¹) when the pore diameter was below $d_n \leq 0.9$ μm (Fig. 4c). With the increasing pore size, the distillation flux of PE membranes goes up from 11.5 to 17.7 Lm⁻²h⁻¹, but the water conductivity rises from 1.1 to 2721 μS cm⁻¹, indicating that the salt rejection drops from 99.99% to 94.96%. The decline in salt rejection performance arises from the low wetting resistance of PE membranes with a large pore size (e.g., $d_n > 1.5$ μm). Considering the standard conductivity of distilled water (0.5–3 μS cm⁻¹), only PE-0.2, PTFE-0.1, and PTFE-0.2 membranes conform to the acceptance criteria.

In contrast, for all the nanofilament-coated PES membranes, the conductivity of purified water remained low and independent of the membrane pore size and distillation flux. The NF-PES-8 membrane performed with an excellent salt rejection (>99.995%) and a substantial increase of distillation flux (18.2 Lm⁻²h⁻¹). Although the surface tension of feed water decreased from 0.07 N m⁻¹ at 50 °C to 0.063 N m⁻¹ at 80 °C, the superhydrophobic nanofilament coating maintained high wetting resistance. Meanwhile, the hierarchical porosity also enhances the thermal efficiency of water production, which is imperative to the future development and industrialization of MD in terms of the water-energy nexus. The thermal efficiency ($\eta$) of tested MD membranes, defined as the ratio of heat utilized for distillation to the total heat consumption at the feed side, is determined by $\eta = q_d/q_f$ (see detailed calculation in Supplementary Note 8). Here, $q_f$ is the total heat transfer

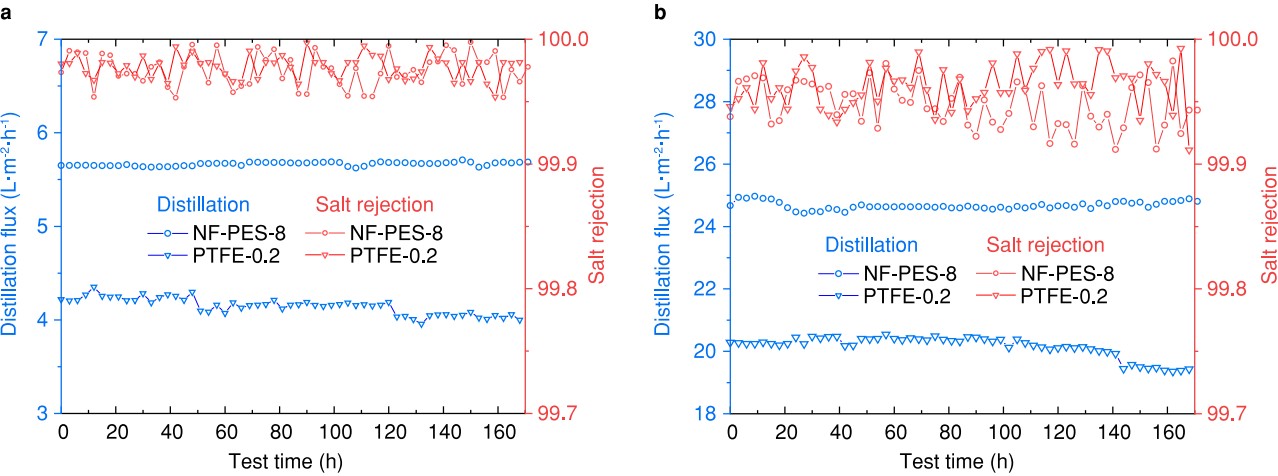

**Fig. 5 | Long-term desalination performance. a** Distillation flux and salt rejection in AGMD operation and **b** in DCMD operation as a function of test time for PTFE-0.2 and NF-PES-8 membranes. The temperatures of feed and cooling water were set at 60 °C and 20 °C to simulate a MD process based on low-grade thermal energy.

rate through the membrane and $q_d$ is vaporization heat transfer rate associated with the distillation flux. As shown in Fig. 4d, the commercial hydrophobic membranes (e.g., PTFE-0.2 and PE-0.2) were able to purify saline continuously, but with a gradual reduction in thermal efficiency from ~86% to ~82% over 12 h. The NF-PES-8 membrane demonstrated a higher and stable thermal efficiency with $\eta$ = ~93% over 12 h under the same AGMD testing condition, significantly outperforming the PE-0.2 and PTFE-0.2 membranes in energy saving.

Direct contact membrane distillation (DCMD) is another MD configuration commonly used in lab research and pilot test. DCMD removes the air gap and condensing surface, so the permeate (i.e., cooling water) directly contacts the membrane. Compared with AGMD, DCMD promises high distillation flux with an advantage in simple system configuration. To further examine the applicability of nanofilament-coated membranes in more practical conditions, we tested the one-week desalination performance of NF-PES-8 and PTFE-0.2 membranes in both AGMD and DCMD configurations. The temperatures of feed and cooling water were set at 60 °C and 20 °C, to simulate the MD process by recovering low-grade thermal energy. Although the low feed temperature decreased the driving force of vapor transport for both AGMD and DCMD, the NF-PES-8 membranes showed a higher distillation flux as compared to the PTFE-0.2 membranes (Fig. 5). In the case of AGMD (Fig. 5a), the NF-PES-8 membrane showed a steady distillation flux of ~5.6 Lm$^{-2}$h$^{-1}$ with salt rejection >99.95% over a whole week. The PTFE-0.2 membrane also exhibited a high salt rejection rate but the MD flux decreased over time from ~4.3 to ~4 Lm$^{-2}$h$^{-1}$, which was 30% lower than that of the NF-PES-8 membrane. In the DCMD test (Fig. 5b) the two membranes demonstrated much higher distillation flux but less stable salt rejection. The absence of the air gap reduced the total mass transfer resistance while increasing the probability of membrane wetting. Nevertheless, the salt rejection of NF-PES-8 membrane remain higher than 99.9%. The one-week DCMD test results showed the same trend in the stability of long-term performance. The distillation flux of NF-PES-8 membrane stabilized at ~24.8 Lm$^{-2}$h$^{-1}$, whereas the value of PTFE-0.2 membrane flux declined from ~20.2 to ~19.2 Lm$^{-2}$h$^{-1}$ when the test time was beyond 100 h.

In order to clearly summarize membrane performance for water desalination, we compared the mass flux coefficient ($\dot{J}$) of the NF-PES membrane with previous studies of AGMD and DCMD, as shown in Fig. 6a and 6b[47–61]. Here, the mass flux coefficient was defined as the ratio of distillation flux to the vapor pressure difference across the membrane ($J/\Delta p$). Due to the insufficient reported data in the literature, we compared the salt rejection of the membranes in AGMD tests

and the LEP of the membranes in DCMD tests. As indicated in Fig. 6a, the NF-PES membrane demonstrates top-tier performance in both mass flux coefficient and salt rejection, which were not achieved simultaneously in previous reports of AGMD tests. Figure 6b shows a similar result that the NF-PES membrane effectively enhances the mass flux coefficient of the DCMD without compromising the LEP. The multi-scale porous structure apparently allows for a better balance between rapid desalination and high salt rejection, which is necessary for commercial MD installation. Moreover, the ultra-high LEP also ensures that the NF-PES membranes are able to withstand extreme hydraulic pressures or purify saline-containing organics (see Supplementary Note 9), suggesting great potential for desalination in very harsh conditions. When comparing our hierarchical membranes to the commercially available PE-0.2 and PTFE-0.1 and PTFE-0.2 membranes (Fig. 6c), the NF-PES-8 membrane shows an up to 60% increased distillation flux. Reducing the pore size of the core membrane to 5 or 3 µm leads to a drop of distillation flux, but increases LEP values. Note that despite relatively lower distillation flux, the NF-PES-0.1 and NF-PES-1.2 membranes still show great potential for desalination in very harsh conditions, where wetting due to contaminants may become limiting. The ultra-high LEP ensures that the membranes are able to withstand extreme hydraulic pressures or are able to purify saline containing low-surface-tension fluids and organics.

## Discussion

In this study, we balanced the conflicting requirements of high wetting resistance and high distillation flux in MD desalination by coating micro-porous membranes with hydrophobic nanofilaments. Without using any fluorine-containing reagents, these composite membranes demonstrated exceptional durability of super liquid-repellency during a prolonged water immersion test. Owing to the multi-scale porous structures, our nanofilament-coated membranes showed superior liquid entry pressure (>11.5 bar), high distillation flux (up to 60% enhancement), and excellent thermal efficiency (>90%) in desalination tests. In one-week AGMD and DCMD tests, they outperform commercial MD membranes currently available (e.g., PE and PTFE). Moreover, the self-assembly-driven fabrication of nanofilament-coated membrane is attractive from the technological perspective, as the procedure is cost-effective (coating cost of $2–$5 m$^{-2}$) and involves no complicated equipment. The superhydrophobic nature of the membrane surfaces leads to improved fouling resistance as found in preliminary tests (Supplementary Fig. 10). One remaining challenge for production scale-up will be the reduction of the reaction time (from an hour timescale to minutes),

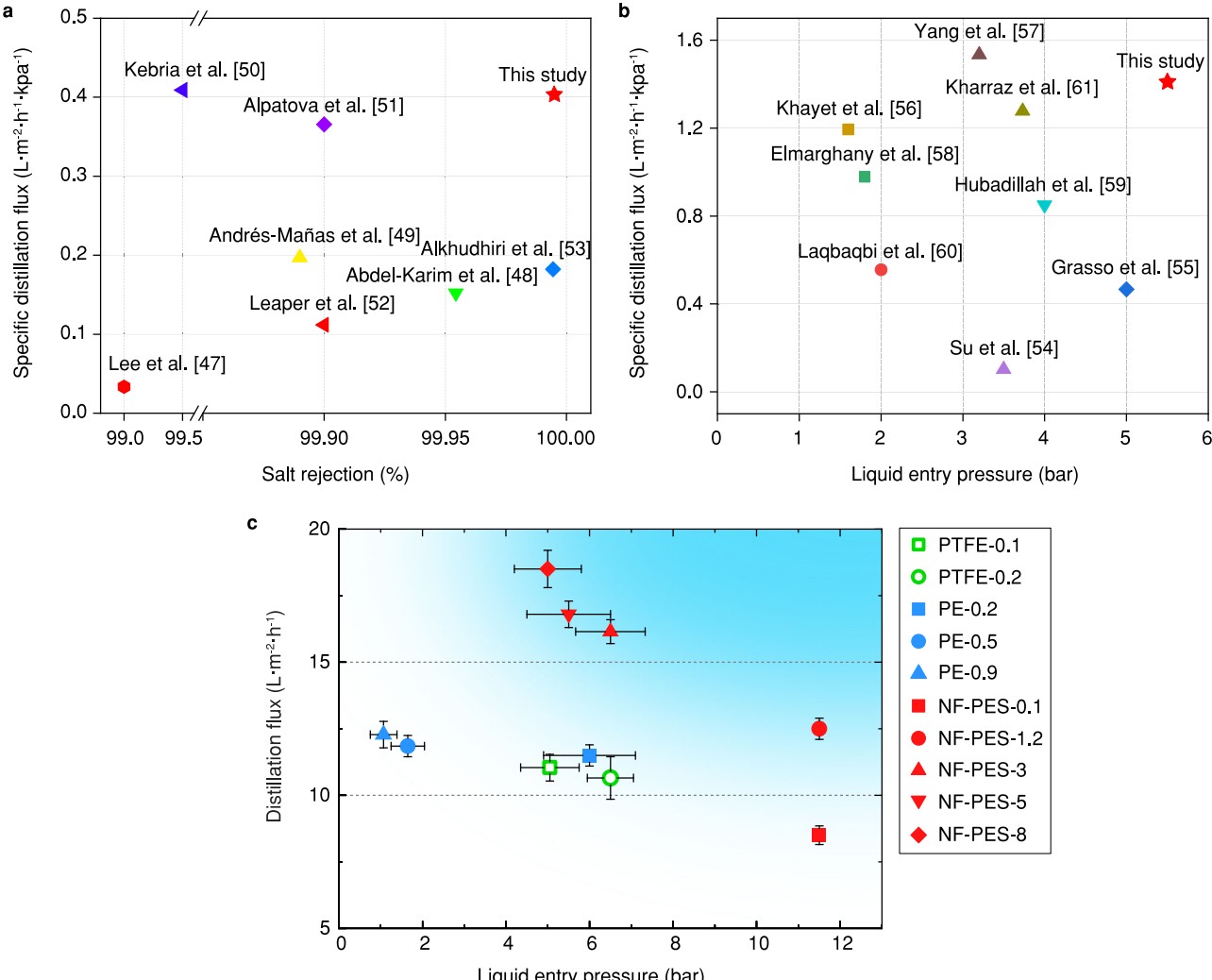

**Fig. 6 | Membrane performance for MD desalination.** Comparison of the **a** AGMD and **b** DCMD performances of the nanofilament-coated membrane (this study) with various previous reports[47–61]. For the AGMD test, the NF-PES membrane can simultaneously achieve high specific distillation flux and high salt rejection. Similarly, the NF-PES membranes effectively enhanced the distillation flux in the DCMD test without compromising the liquid entry pressure (LEP). **c** Comparison of key properties of nanofilament-coated PES membranes (red symbols) and commercially available membranes (green and blue symbols). An ideal MD membrane for water desalination should meet the simultaneous requirements of high LEP and high distillation flux (as suggested by blue area in upper right sector). Error bars indicate standard deviation ($n = 5$).

which might be feasible by further optimizing reaction conditions. The outcome of this work will not only solve the fundamental issues that have persisted in the MD process, but will also pave the way to using advanced hierarchical porous membranes in a broader range of water treatment applications[62,63].

## Methods

### Fabrication of the nanofilament-coated membranes

PES membranes, purchased from Sterlitech Corp., USA, were activated by using $O_2$ plasma (2 min, 90 W, Diener Electronic Femto). The $O_2$ flow rate was set to 7 ml min$^{-1}$. 0.6 ml of trichloromethylsilane (TCMS, 99% purity) was added to 300 ml of a 1:1 (volumetric) mixture of $n$-heptane (99% purity) and toluene (99.8% purity). Before mixing, a trace amount of MilliQ water was added to $n$-heptane sand toluene. The water concentration of $n$-heptane and toluene was measured as 90 ppm and 275 ppm, respectively. Then the plasma activated PES membranes were immersed in the reaction solution for nanofilament growth, which occurs then by spontaneous 1D growth originating from –OH groups at the membrane surface by hydrolysis of TCMS. After 6 h, the nanofilament-coated PES membranes were rinsed with $n$-hexane and dried using $N_2$ flow.

### Fluorination of hydrophilic PES membranes

For comparison, the originally hydrophilic PES membranes were hydrophobized by using surface fluorination with 1H,1H,2H,2H-perfluorodecyltrichlorosilane (PFDTS, 96%). PES membranes were activated using $O_2$ plasma treatment (2 min, 90 W, Diener Electronic Femto) at a $O_2$ flow rate of 7 ml min$^{-1}$. Subsequently, PFDTS (180 μL, 96% purity) was mixed with $n$-hexane (350 mL, 95% purity) and the activated membranes were immersed in the solution for 60 min, rinsed with $n$-hexane and dried under a nitrogen gas flow.

### Contact angle measurements

The water contact angles of all membranes (PE, PTFE, fluorinated PES and nanofilament-coated PES membranes) were measured to characterize the surface wettability. Contact angle and contact angle hysteresis of a water droplet were measured using a DataPhysics OCA35 goniometer. During the measurement, a 5 μl droplet was deposited on the membrane surface, and afterward 20 μl of water was added to and then removed from the droplet. The measurement was consecutively repeated three times at the same position, and at three different positions per substrate. The error of the advancing and receding contact angle measurements was estimated to be ±2°.

## Liquid entry pressure (LEP) measurements

To measure the LEP of membranes, we built a custom-designed apparatus (see Supplementary Fig. 6 and Supplementary Note 5). The tested membrane was mounted inside a filter holder, which connected to a syringe pump. By slowly pumping the salty water into the filter holder (0.1 mL min⁻¹), the hydrostatic pressure applied on the tested membrane gradually increased. The hydrostatic pressure was monitored using a pressure sensor (IPSLU-M12, RS-Pro) and the data was recorded using a data acquisition system (PCI 6251, National Instruments). Once the applied pressure exceeded the capillary pressure of the membrane pores, liquid penetrated the membranes, leading to a pressure drop. The obtained peak value of the pressure measurement gives the LEP of the tested membrane.

## Gas permeability measurements

A gas permeability test was employed to analyze the mass transfer resistance of different membranes (see Supplementary Figs. 7 and 8, and Supplementary Note 6). The permeation flux of nitrogen through the dry membranes was measured under transmembrane pressures ranging from 10 to 1000 mbar. With increasing transmembrane pressure, the gas flow rate was obtained by using flow sensors with respective ranges (SMC Corp., PFMV5 series). The effective area of the tested membrane was 63 mm².

## Membrane distillation tests

Membrane distillation tests were performed using a custom-made AGMD setup, which consisted of AGMD module, feed water and coolant circulating loops, digital balance, conductivity meter, and data acquisition system (see Supplementary Fig. 9). The tested membrane was mounted in the AGMD module, between a feed flow channel and a condensing surface. A support mesh (~0.5 mm thick) was used to hold the membrane in a planar shape and reduce the membrane deformation due to the pressure difference between feed flow and air gap. An acrylic spacer was used in the MD module to create the required air gap. The total air gap width between membrane and condensing surface was ~4.5 mm. Feed saline water was heated to the desired temperature and pumped to the AGMD module using a magnetic coupling water pump. The condensing surface temperature was controlled by the coolant flow loop using a refrigerated water bath circulator (F25-HE, Julabo). When distilled water slid off the condensing surface by gravity, it was collected in a glass flask. A digital balance (SPX 2202, Ohaus) continuously recorded the weight of collected distilled water for determining the distillation flux of tested membranes. The conductivities of feed and distilled water were measured by the conductivity meter for calculating the salt rejection during membrane distillation. Four Pt100 temperature probes (PM-1/10-1/8-6-0-P-3, Omega) were adopted to measure the liquid temperature at inlet and outlet of feed flow channel and coolant flow channel, respectively. Two flow meters (FT110, Gems) and two pressure transducers (IPSLU-M12, RS-Pro) were installed in the pipelines to continuously monitor the flow rate and pressure in the feed and coolant loops. All the sensors in the AGMD testing setup were electrically connected to a data acquisition system, which consisted of two National Instruments (NI) analog input modules (PCI 6251 and NI-9216). The measured data during MD experiments were transferred to the computer, which could be monitored in real-time and stored using a self-written LabView code.

## Data availability

The data that supports the findings of this study are provided in the main text and supplementary information files. Source Data file has been deposited in Figshare under accession code DOI link https://doi.org/10.6084/m9.figshare.23958075.

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

## Acknowledgements

This project has received funding from the European Union's Horizon 2020 research and innovation programme under grant number 801229 HARMoNIC (Y.H., P.S., V.C., E.G., M.K. H-J.B.). Y.H. also acknowledges the funding support by the MSCA Individual Fellowship (TrapJump 895899), the Humboldt Research Fellowship for Postdoctoral Researchers, and the National Natural Science Foundation of China (No. 52276160).

## Author contributions

Y.H., E.G., M.K., and H.B. conceived the project. Y.H., P.S., and V.C. carried out the fabrication, experiments, and characterization. Y.H., P.S., V.C., and M.K. worked on the analysis of experimental results. Y.H., M.K., E.G., and H.B. wrote the manuscript. All authors discussed the results and commented on the manuscript.

## Funding

## Competing interests

The authors declare no competing interests.
