## [Peer review file · Nature Communications]

REVIEWERS' COMMENTS

Reviewer #1 (Remarks to the Author):

The main idea here is that by making a thin superhydrophobic coating on a microporous substrate with large pores, one can construct a high-performance membrane for membrane distillation (MD). The microporous substrate has large pores that allow for fast vapor transfer but suffers from low liquid entry pressure (LEP), the nano filament coating increases the LEP and addresses the low-LEP challenge, but is thin enough not to induce too much vapor transport resistance. The authors argue that such a design can lead to a high-performance MD membrane over commercial available membranes.

I found this design to be logical and elegant. Although compared to the state-of-the-art membranes (commercial or lab-fabricated), the membranes reported here are not really superb in terms of flux or permeability, I still believe the work carries the merits for publication in Nature Comm.

The manuscript is well written with clarity. I only have one major question. The nanofilament coating seems to be quite thin with pores that are not sufficiently small. With this aspect ratio, wetting could be a challenge based on the "grand potential" analysis (not just LEP based on force balance). From the experimental data it seems wetting is not a concern. I would suggest the author to analyze the wetting propensity based on the grand potential approach following this paper just to make sure.

Restagno, F., Bocquet, L. and Biben, T., 2000. Metastability and nucleation in capillary condensation. Physical review letters, 84(11), p.2433.

Reviewer #2 (Remarks to the Author):

All comments fully addressed. No further comments and I recommend the manuscript to be published.

Response to referee #1

We thank the referee for pointing out the possible issue of capillary condensation. The theoretical modelling in the mentioned PRL paper refers to the case of hydrophilic surfaces. Since our nanofilament layer is superhydrophobic, capillary condensation is not likely to occur at all. If capillary condensation would occur inside the nanofilament layer, we would expect a gradual decrease in distillation flux. However, even for one week of continuous operation, stable distillation flux is observed. In addition, to carry out an analysis of the wetting propensity based on a grand canonical approach, the model would have to be adjusted to this different wetting situation (superhydrophobic vs. hydrophilic), which is beyond the scope of our experimental study.

We have therefore added the following paragraph to the manuscript including citation of the paper by Restagno et al.:

Besides membrane wetting by imbibition, as characterized by LEP and described theoretically by the Young-Laplace equation, capillary condensation inside of membrane pores could in principle be another mechanism of wetting. Such a wetting process could be described theoretically by a grand potential approach for hydrophilic surfaces³³. However, the micrometer-sized pores of the hydrophilic core PES membrane are too large to lead to capillary condensation. Due to the hydrophobic character of the nanofilaments, capillary condensation is not expected to occur within the coating layer. If capillary condensation would occur within the nanofilament layer, it would lead to a gradual decrease of the distillation flux, which we do not observe even during prolonged distillation tests.

33 Restagno F, Bocquet L, Biben T. Metastability and Nucleation in Capillary Condensation. *Phys. Rev. Lett.* **84**, 2433-2436 (2000).